# A comprehensive risk score for effective risk stratification and screening of nasopharyngeal carcinoma

Xiang Zhou[1,12], Su-Mei Cao[1,12], Yong-Lin Cai[2,12], Xiao Zhang[1,12], Shanshan Zhang[1], Guo-Fei Feng[3], Yufeng Chen[4], Qi-Sheng Feng[1], Yijun Chen[1], Ellen T. Chang[5,6], Zhonghua Liu[7], Hans-Olov Adami[4,8], Jianjun Liu[9,10], Weimin Ye[4,11], Zhe Zhang[3✉], Yi-Xin Zeng[1✉] & Miao Xu[1✉]

Using Epstein-Barr virus (EBV)-based markers to screen populations at high risk for nasopharyngeal carcinoma (NPC) is an attractive preventive approach. Here, we develop a comprehensive risk score (CRS) that combines risk effects of EBV and human genetics for NPC risk stratification and validate this CRS within an independent, population-based dataset. Comparing the top decile with the bottom quintile of CRSs, the odds ratio of developing NPC is 21 (95% confidence interval: 12–37) in the validation dataset. When combining the top quintile of CRS with EBV serology tests currently used for NPC screening in southern China, the positive prediction value of screening increases from 4.70% (serology test alone) to 43.24% (CRS plus serology test). By identifying individuals at a monogenic level of NPC risk, this CRS approach provides opportunities for personalized risk prediction and population screening in endemic areas for the early diagnosis and secondary prevention of NPC.

[1] State Key Laboratory of Oncology in South China, Collaborative Innovation Center for Cancer Medicine, Guangdong Key Laboratory of Nasopharyngeal Carcinoma Diagnosis and Therapy, Sun Yat-sen University Cancer Center, Guangzhou, P. R. China. [2] Department of Clinical Laboratory, Wuzhou Red Cross Hospital, Wuzhou, China. [3] Department of Otolaryngology/Head and Neck Surgery, First Affiliated Hospital of Guangxi Medical University, Nanning, China. [4] Department of Medical Epidemiology and Biostatistics, Karolinska Institutet, Stockholm, Sweden. [5] Center for Health Sciences, Exponent, Menlo Park, CA, USA. [6] Stanford Cancer Institute, Stanford, CA, USA. [7] Department of Statistics and Actuarial Science, The University of Hong Kong, Hong Kong SAR, China. [8] Clinical Effectiveness Group, Institute of Health and Society, University of Oslo, Oslo, Norway. [9] Human Genetics, Genome Institute of Singapore, Agency for Science, Technology and Research (A*STAR), Singapore, Singapore. [10] Department of Medicine, Yong Loo Lin School of Medicine, National University of Singapore, Singapore, Singapore. [11] Department of Epidemiology and Health Statistics & Key Laboratory of Ministry of Education for Gastrointestinal Cancer, Fujian Medical University, Fuzhou, China. [12] These authors contributed equally: Xiang Zhou, Su-Mei Cao, Yong-Lin Cai, Xiao Zhang. ✉email: zhangzhe@gxmu.edu.cn; zengyx@sysucc.org.cn; xumiao@sysucc.org.cn

Nasopharyngeal carcinoma (NPC) is causally related to infection with Epstein–Barr virus (EBV). In most of the world, NPC has an incidence below one per 100,000 person-years, but it is endemic in southern China, Southeast Asia, North Africa, and the Arctic[1]. The incidence of NPC is particularly high, at 20–40 per 100,000 person-years, in southern China[2]. Screening is the main secondary prevention strategy for NPC in endemic areas because no prophylactic vaccine against EBV is available. Although EBV infection is ubiquitous worldwide, the endemicity of NPC in southern China might be largely attributable to specific oncogenic EBV subtypes[3]. Other risk factors for NPC include male sex, ages 40–50 years, lower socioeconomic status, family history of NPC, genetic susceptibility, and certain lifestyle factors[4,5]. Several genome-wide association studies (GWASs) have confirmed the association of the *HLA, TERT, CDKN2A/2B, MECOM,* and *TNFRSF19* gene loci with NPC susceptibility[6–8]. Among lifestyle factors, consumption of salt-preserved fish and smoking are most consistently associated with NPC[9].

Strong associations of elevated EBV serological markers and plasma EBV DNA levels with NPC risk have been consistently observed in populations across different endemic areas[10,11]. Several EBV-based biomarkers have been proposed for screening in high-risk populations. Of these markers, IgA antibodies against viral capsid antigens (VCA-IgA) and nuclear antigen 1 (EBNA1-IgA) have been used in several NPC screening trials for the general population in southern China[12–14]. However, despite high sensitivity and specificity of these markers, EBV serology tests are limited by their low positive predictive value (PPV), largely due to the rarity of NPC even in endemic areas[14]. To improve PPV, other approaches to identify high-risk individuals and prioritize them for EBV serology or early diagnostic testing (e.g., using plasma EBV DNA[10]) should be considered. Based on our recent finding that newly identified EBV subtypes can explain more than 80% of the population risk of NPC[3], we hypothesized that incorporating EBV high-risk genetic variants (at positions 162215, 162476, and 163364) and key epidemiological risk factors along with human genetic variants could improve screening effectiveness.

In this study, utilizing information on EBV genetic variants, host genetic susceptibility, and epidemiological risk factors for NPC in a subset of participants (892 cases and 1340 controls) from a large, population-based NPC case-control study in southern China, we developed and validated a comprehensive risk score (CRS). We showed that the odds ratio of developing NPC was approximately 21 (95% CI: 12–37) among individuals in the top decile of CRS, compared with individuals in the bottom quintile. By combining the CRS with EBV serology tests currently used for NPC screening, the PPV of screening increased from 4.70% (serology alone) to 43.24% (CRS plus serology). Thus, our study demonstrates the high potential of CRS for NPC risk discrimination and stratification in southern China.

## Results

### Characteristics of the study participants and the associations of risk factors with NPC.
The study design is outlined in Fig. 1. Briefly, we genotyped saliva DNA from 1710 cases and 2246 controls for three EBV variants at positions 162215, 162476, and 163364, and seven human SNPs associated with NPC risk from previously reported GWASs. Genotyping information on human SNPs was obtained from 96.7% (3826/3956) of participants, while EBV variant information on all three positions was available for 955 cases (55.8%) and 1423 controls (63.4%). The success rate for EBV genotyping was most likely determined by the quantity of EBV DNA in saliva, since EBV is intermittently shed in the oropharynx of normal adults. To evaluate potential bias, we compared the characteristics of individuals with successful EBV genotyping to those without. In our dataset, the missingness of EBV genotyping data did not differ by age, salted-fish consumption, educational level, or a family history of NPC (Supplementary Fig. 1a). A lower frequency of missing EBV genotyping was found among smokers and men (who are substantially more likely than women to be smokers) (Supplementary Fig. 1b). This pattern is concordant with the observation that smoking stimulates EBV lytic production[15,16], which increases the chances of EBV being genotyped. However, because the relative risk and attributable risk of NPC associated with smoking are relatively small, with a relative risk of only 1.1–1.5, any selection bias caused by overinclusion of smokers in our dataset would be small. Additionally, smoking status and educational level were included as covariates in multivariate models (and cases and controls were frequency matched by age and sex) to adjust for potential confounding effect in this study.

Characteristics of participants with information on all three EBV SNPs, seven human SNPs and the covariates (sex, age, family history of NPC, salted-fish consumption, smoking and educational level) are shown in Table 1. The training and validation datasets had similar distributions by case-control status, sex, and age, reflecting the frequency-matched study design. After considering potential epidemiological risk factors for NPC, e.g., family history of NPC, salted-fish consumption, smoking, socioeconomic status, educational level, tea drinking, wood dust exposure, formaldehyde exposure, and oral hygiene, we included four factors in the CRS: family history of NPC, salted-fish consumption, smoking, and educational level (Table 1). These are well-defined risk factors with consistent epidemiological evidence, and are relatively straightforward for individuals to recall and quantify. Among the seven reported human SNPs associated with NPC in GWASs, we confirmed two SNPs, rs2860580 and rs2894207, in the *HLA* region that were significantly associated with NPC in both datasets. Statistically nonsignificant positive associations were observed for the other five human SNPs (Supplementary Table 1). For the three EBV SNPs (162215, 162476, and 163364), we confirmed strong associations with NPC risk in both the training and validation datasets (Supplementary Table 1).

### Comprehensive risk score (CRS).
Based on the training dataset, we built three logistic regression models for NPC risk: model #1, including four well-established epidemiological risk factors (smoking status, salted-fish consumption, educational level, and family history of NPC); model #2, including the four epidemiological risk factors and two human *HLA* SNPs (rs2860580 and rs2894207); and model #3 (the comprehensive model), including the same factors as model #2 plus the three EBV genetic variants. In the validation set, the comprehensive model displayed substantially and statistically significant better discrimination between cases and controls (areas under the curve [AUC] = 0.772, 95% confidence interval [CI]: 0.745–0.800) than the other two models (all *P*-values < 1.00E-22, Table 2, Fig. 2a; Training dataset: Supplementary Table 2 and Supplementary Fig. 2a). The EBV variants contributed the most to prediction performance, increasing AUC from 0.619 to 0.772 in the validation dataset (Table 2, Training dataset: Supplementary Table 2). The two human SNPs (AUC = 0.761, *P* = 1.20E−02) and the epidemiological factors (AUC = 0.764, *P* = 2.87E-03) also made small but statistically significant contributions to model performance in the validation dataset (Supplementary Table 3). The prediction increments of the two human SNPs (21%, 95% CI: 15–27%) and the epidemiological factors (9%, 95% CI: 5–12%) in the validation

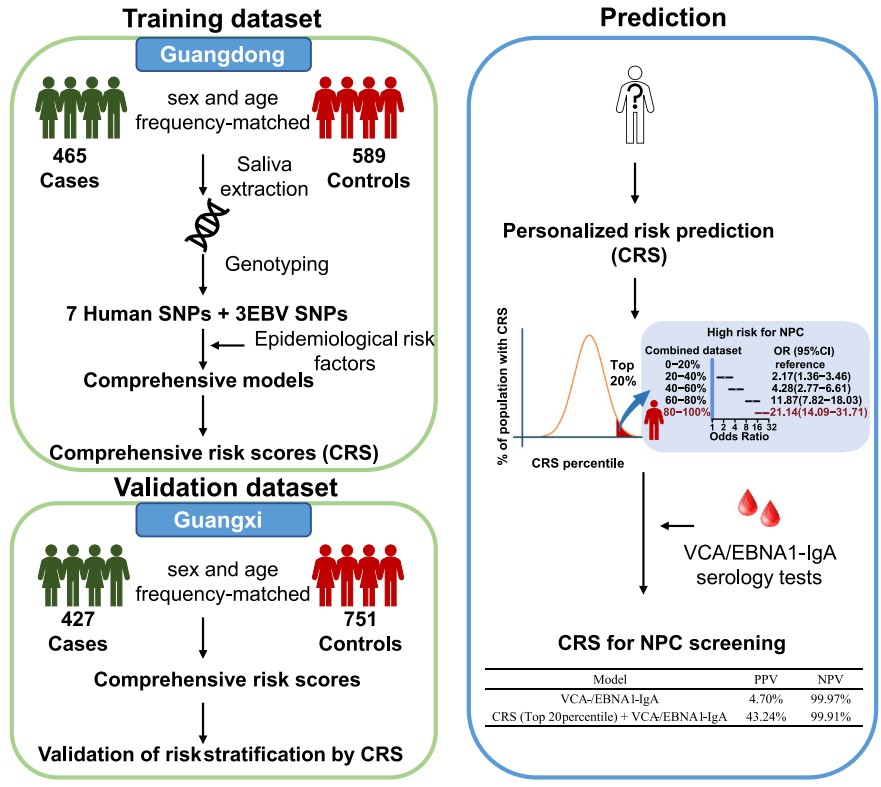

**Fig. 1 Workflow of comprehensive risk score (CRS) construction and its application in NPC screening.** The CRS was developed using EBV, host genetic and epidemiological risk factors for NPC risk prediction with a discovery dataset from Guangdong province, and validated in an independent dataset from Guangxi province. The CRS was combined with EBV serology tests for NPC screening.

| Table 1 Demographic characteristics of the study participants in the training and validation datasets. | | | | | | |
|---|---|---|---|---|---|---|
| **Variable** | **Training dataset** | | | **Validation dataset** | | |
| | **Cases (%)** | **Controls (%)** | **P value[a]** | **Cases (%)** | **Controls (%)** | **P value[a]** |
| *Age* | | | | | | |
| Mean (SD) | 48.38 (11.07) | 49.22 (10.64) | 2.15E−01 | 49.30 (10.83) | 50.98 (10.91) | 1.07E−02 |
| < 28 | 16 (3.44) | 13 (2.21) | 4.23E−01 | 7 (1.64) | 10 (1.33) | 1.06E−02 |
| 28–59 | 377 (81.08) | 477 (80.98) | | 347 (81.26) | 556 (74.03) | |
| > 59 | 72 (15.48) | 99 (16.81) | | 73 (17.10) | 185 (24.63) | |
| *Sex* | | | | | | |
| Men | 349 (75.05) | 453 (76.91) | 5.29E−01 | 330 (77.28) | 581 (77.36) | 1.00E+00 |
| Women | 116 (24.95) | 136 (23.09) | | 97 (22.72) | 170 (22.64) | |
| *Smoking* | | | | | | |
| Never | 174 (37.42) | 217 (36.84) | 7.59E−01 | 182 (42.62) | 335 (44.61) | 3.79E−01 |
| Former[b] | 34 (7.31) | 37 (6.28) | | 27 (6.32) | 34 (4.53) | |
| Current | 257 (55.27) | 335 (56.88) | | 218 (51.05) | 382 (50.87) | |
| *Salted fish* | | | | | | |
| Yearly or less | 314 (67.53) | 381 (64.69) | 5.84E−02 | 373 (87.35) | 612 (81.49) | 1.62E−02 |
| Monthly | 102 (21.94) | 162 (27.50) | | 48 (11.24) | 113 (15.05) | |
| weekly or more | 49 (10.54) | 46 (7.81) | | 6 (1.41) | 26 (3.46) | |
| *Education* | | | | | | |
| Illiterate or primary school | 183 (39.35) | 205 (34.80) | 2.55E−01 | 183 (42.86) | 284 (37.82) | 2.22E−01 |
| Middle school | 192 (41.29) | 244 (41.43) | | 156 (36.53) | 292 (38.88) | |
| High school | 70 (15.05) | 113 (19.19) | | 73 (17.10) | 134 (17.84) | |
| College or higher | 20 (4.30) | 27 (4.58) | | 15 (3.51) | 41 (5.46) | |
| *Family history* | | | | | | |
| No | 406 (87.31) | 568 (96.43) | 5.46E−08 | 389 (91.10) | 732 (97.47) | 1.97E−06 |
| Yes | 59 (12.69) | 21 (3.57) | | 38 (8.90) | 19 (2.53) | |

[a] The P value was calculated using a two-sided *t*-test. Other P values for categorical variables were calculated using $\chi^2$ tests.
[b] Individuals who had quit smoking more than 1 year before the interview were defined as former smokers.

**Table 2 The performance of the models for distinguishing the patients with NPC from the controls.**

| Model | Validation dataset | | | | | |
| --- | --- | --- | --- | --- | --- | --- |
| | AUC | 95%CI | Repeated 10-fold | P value[a] | P value[a] | R[2b] |
| Epidemiology model [c] | 0.563 | 0.529–0.597 | 0.586 | 9.03E-26 | ------[d] | 0.016 |
| Epidemiology + 2 host SNPs model[e] | 0.619 | 0.585–0.652 | 0.631 | 9.37E-23 | 1.85E-03 | 0.031 |
| Epidemiology + 2 host SNPs + 3 EBV SNPs model[f] | 0.772 | 0.745–0.800 | 0.770 | ------[d] | 9.03E-26 | 0.174 |

*AUC* area under the curve, *CI* confidence interval, repeated 10-fold, average AUC from repeated 10-fold cross-validation.
[a] The P value was calculated using a two-sided Delong's test.
[b] McFadden's Pseudo-$R^2$.
[c] Epidemiological model: smoking, salted-fish consumption, education and family history of NPC.
[d] "–" is a reference model.
[e] 2 Host SNPs: rs2860580, rs2894207.
[f] 3 EBV SNPs: EBV162215, EBV162476 and EBV163364.

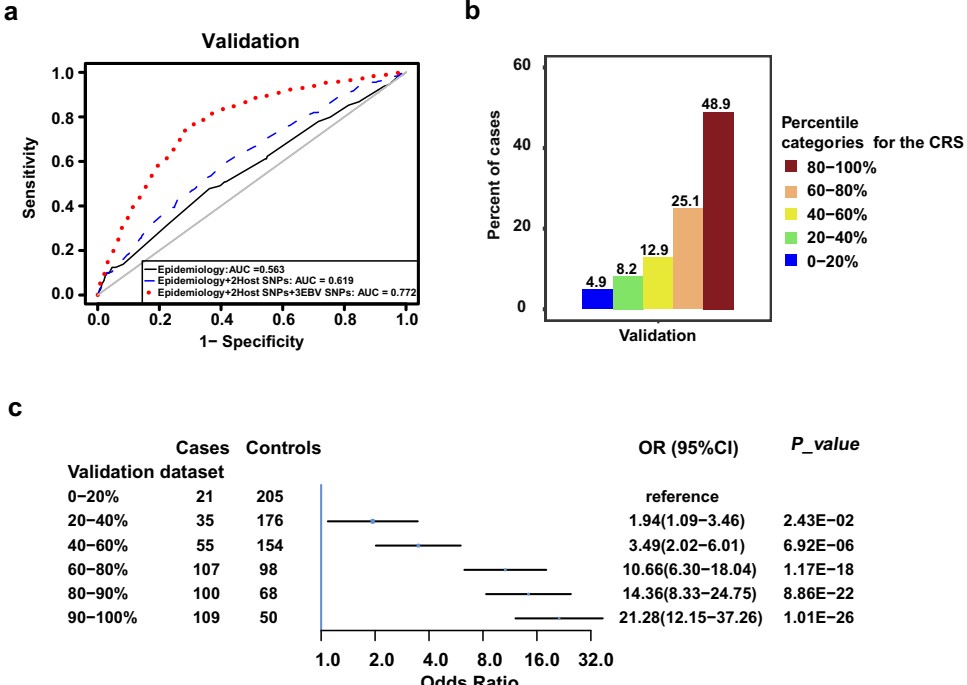

**Fig. 2 NPC risk stratified by comprehensive risk score (CRS). a** Receiver operating characteristic (ROC) curve analysis of the validation dataset. The area under the ROC curve (AUC) for each model is indicated. **b** Distribution of patients in the validation dataset stratified into five categories, 0–20%, 20–40%, 40–60%, 60–80%, and 80–100%, according to the CRS percentile among controls in the training dataset. **c** Association between the CRS and NPC risk in the validation dataset (Cases: n = 427, Controls: n = 751). The validation dataset was stratified into categories (lower four quintiles and top two deciles) according to the CRS percentile among controls in the training dataset, and participants in the bottom quintile of CRS served as the reference group. The odds ratio (OR) of developing NPC was estimated using logistic regression analysis with each group included as a categorical variable. The blue squares represent the odds ratios of each category, and the error bars represent the 95% confidence intervals. CI, confidence interval. Source data of (**b**) are provided in the Source Data file.

dataset were also confirmed by the net reclassification index (Supplementary Table 4). The comprehensive model showed good internal validity and little evidence of overfitting (average AUC from repeated 10-fold cross-validation = 0.770; Table 2). Hosmer-Lemeshow plots indicated a better calibration of the comprehensive model than the other two models (Nagelkerke pseudo-$R^2$ = 0.23, P value of Pearson chi-squared goodness-of-fit test = 0.68, Supplementary Fig. 2b).

We therefore constructed the comprehensive risk score (CRS) based on the comprehensive model using the training dataset. NPC cases had higher CRSs (mean = 3.64, SD = 1.05) than controls (mean = 2.43, SD = 1.21) in the validation dataset (Supplementary Fig. 3a). In quintiles categorized based on the distribution of the CRS among controls in the training dataset, 48.9% of NPC cases in the validation dataset had a CRS in the top

quintile, whereas only 4.9% of cases had a CRS in the bottom quintile (Fig. 2b, Training dataset: Supplementary Fig. 3b). In the validation dataset, we found a strong, monotonic positive trend between the CRS and NPC risk (Fig. 2c, Training dataset: Supplementary Fig. 3c). Individuals in the ninth decile of CRS had a greater than 14-fold higher risk (95% CI: 8.33–24.75) of developing NPC than individuals in bottom quintile, and those in the tenth (highest) decile had a 21-fold higher risk (95% CI: 12.15–37.26, Fig. 2c).

**Adding the CRS to standard serum EBV VCA-/EBNA1-IgA screening tests enhances identification of individuals at high risk of developing NPC.** To evaluate the potential of CRS for NPC screening, we combined the CRS with serum EBV VCA-/

**Table 3 Prediction accuracy of the serum VCA-/EBNA1-IgA antibody tests and the models combining serum EBV antibody levels and the comprehensive risk score with different cutoff values in the combined dataset.**

| Model | Cut off | Sensitivity | Specificity | PPV | NPV |
|---|---|---|---|---|---|
| VCA-/EBNA1-IgA | $P \geq 0.98$ | 80.13% | 97.39% | 4.70% | 99.97% |
| CRS (Top 40 percentile) + VCA-/EBNA1-IgA | CRS $\geq 3.23$ and $P \geq 0.98$ | 62.95% | 99.48% | 16.22% | 99.94% |
| CRS (Top 30 percentile) + VCA-/EBNA1-IgA | CRS $\geq 3.62$ and $P \geq 0.98$ | 53.21% | 99.83% | 32.93% | 99.92% |
| CRS (Top 20 percentile) + VCA-/EBNA1-IgA | CRS $\geq 3.88$ and $P \geq 0.98$ | 41.28% | 99.91% | 43.24% | 99.91% |
| CRS (Top 10 percentile) + VCA-/EBNA1-IgA | CRS $\geq 4.30$ and $P \geq 0.98$ | 25.64% | 99.91% | 32.12% | 99.88% |

CRS comprehensive risk score, AUC area under the curve, CI confidence interval, PPV positive predictive value, NPV negative predictive value.

EBNA1-IgA tests for discriminating NPC cases, and estimated the PPVs and NPVs based on the test specificity, sensitivity, and NPC prevalence in southern China (Methods). We combined the CRS at various cutoff values with EBV serology test results, and classified only individuals who were identified as high-risk by both CRS and serology tests as the high-risk group. In the combined dataset, the PPV of the combined tests (43.24%), using the top quintile of CRS as a cutoff and positivity for VCA-/EBNA1-IgA, was 9-fold greater than the PPV using EBV serology alone (4.70%), with a subtle decrease in the NPV (99.91%) (Table 3). Using a looser CRS cutoff at the top 30%, combined with positive EBV serology, achieved a specificity of 100%, resulting in the highest PPV in the validation dataset (Supplementary Table 5). Setting the cutoff at the top decile of CRS did not further increase the PPV, because the sensitivity decreased without a further gain in specificity (Table 3 and Supplementary Table 5). These results indicate that implementing the CRS as an addition to standard EBV VCA-/EBNA1-IgA serology tests substantially augments the PPV for discriminating NPC in an endemic southern Chinese population.

## Discussion

Based on a large, population-based case-control study in NPC-endemic southern China, we report the development and validation of a CRS that combines EBV and human genetic factors with key epidemiological risk factors to identify individuals at monogenic level of NPC risk. The odds ratio of developing NPC was approximately 21 (95% CI: 12–37) among individuals in the top decile of CRS compared with individuals in the bottom quintile. By combining the CRS with EBV serology tests currently used for NPC screening in southern China, and setting the CRS cutoff at the top quintile among population controls, we observed a remarkable increase in the PPV for NPC screening, from 4.70% to 43.24%. This combined method thus shows great potential to enhance population screening for NPC.

NPC prognosis depends strongly on the extent of disease at diagnosis. Patients at stage I have an excellent long-term prognosis, with five-year survival above 90%, whereas patients with advanced NPC (stages III/IV) harbor a substantial risk of distant metastatic relapse (~ 20–30%)[17]. Unfortunately, since early-stage NPC usually shows no or mild signs and symptoms, fewer than 20% of NPC patients in endemic regions of China are diagnosed with early stage tumors. Screening for high-risk individuals is currently the main NPC prevention strategy. Based on strong associations of EBV serology and plasma EBV DNA with NPC risk, these markers have been suggested for NPC screening programs in mainland China, Taiwan, Southeast Asia and Hong Kong. However, EBV serology tests have a PPV of only approximately 1–5%[11,13,18–20]. Although PPV was markedly higher (11–20%) when screening for plasma EBV DNA among Hong Kong men aged 40–62 years, the age interval when NPC risk peaks[10,21], the PPV among men and women across all age groups would be expected to be much lower due to that NPC

incidence among women is about 1/3 of that among men. Because plasma EBV DNA is derived largely from apoptosis or secretion of NPC cells[22,23], this biomarker is anticipated to be more suitable for early diagnosis and prognostic monitoring, rather than risk prediction prior to tumor development. A novel screening strategy with higher PPV thus remains a high public health priority for the general population in NPC-endemic regions.

By explaining more than 80% of the overall risk for NPC in southern China, the recently discovered NPC-associated EBV risk subtypes have excellent potential as biomarkers for advancing population screening[3]. Based on a small number of samples, a recent study suggested that the addition of EBV genetic variants can also improve the PPV of NPC screening based on plasma EBV DNA load[24], supporting the notion that combining a broad panel of EBV markers could improve population screening. In the present study, although EBV genetic variants were responsible for the majority of NPC risk prediction, we observed a statistically significant improvement in risk stratification based on the CRS by also incorporating host genetic risk factors verified by several GWASs and key epidemiological risk factors. The top quintile of the CRS successfully identified individuals at a monogenic level of increased NPC risk in the independent validation dataset, demonstrating its high potential for clinical application. The subpopulation with a high CRS thus appears to constitute a high-risk group worthy of further screening (e.g., with repeated EBV serology) or early diagnostic testing (e.g., based on MRI, nasal endoscopy, or plasma EBV DNA).

EBV infection occurs early in life in NPC-endemic populations, long before the risk of NPC peaks at 40–50 years of age[25,26]. The CRS, therefore, can potentially identify individuals at high risk of NPC as early as EBV has established lifelong latency. When we combined standard EBV serology test results with the CRS, setting the cutoff at the top quintile in the general population (represented by population-based controls), the PPV improved by 9-fold (to 43.24%) compared with EBV serology alone (4.70%). Our results thus suggest that the CRS could be used as the first step of population screening to identify individuals at the highest risk of NPC, with this group proceeding to the second step of NPC screening based on EBV serology tests. In the general population from endemic areas of southern China, approximately 1–5% test at high risk with EBV VCA-/EBNA-1-IgA serology test[12]. Because the CRS was not correlated with EBV serology (Supplementary Fig. 4), the CRS appears to be an independent, nonduplicative predictor of NPC risk. Accordingly, we estimate that approximately 1–5% of the population in the top quintile of the CRS would also test at high risk for EBV serology. Hence, using this two-step screening strategy, 0.2–1% of the general population would be stratified into the highest-risk group by both the CRS and EBV serology tests, and could potentially benefit from further preventive actions or surveillance for early diagnosis at an appropriate age. The current cost of human and EBV genotyping, as performed in our study, is approximately $16 in

U.S. dollars; additionally, collecting self-reported epidemiological risk factor data would contribute a negligible cost. Considering the relatively low cost and high PPV, the CRS combined with EBV serology test thus provides a promising and cost-effective tool for risk discrimination that may aid in clinical decision-making and resource allocation.

This study has some limitations. The CRS generated in this study was developed and validated in a southern Chinese population with high NPC incidence. As shown in a publication, the distribution of these three risk EBV variants among 20 NPC cases and 4 controls recruited from Indonesia[27] is quite different from that in southern China (Supplementary Table 6). Despite the small sample size, their evidence might highlight that our CRS model could be population specific. As this CRS approach has the potential to be an important screening tool, it is important to assess the performance in early-stage tumors. However, because Stage I and II NPC tumors were minority of the patient dataset in this study (Supplementary Table 7), we could not evaluate the performance of this CRS approach to identify early-stage NPC. Nevertheless, this study offers a compelling proof of concept that using combined EBV and host genotype data and epidemiological risk factor data to predict NPC risk can strongly augment the efficacy of NPC screening based on EBV serology.

In addition, the CRS can be calculated only in those with EBV genotype information available (based on our study, approximately 60% of the population). Reassuringly, we found that missing saliva EBV genotyping data was not associated with increased risk of NPC, and it was unrelated to the majority of NPC risk factors, including sex (which was explained by smoking), age, salted-fish consumption, educational level, family history of NPC and VCA/EBNA1-IgA in our data. The population-based controls were frequency-matched to cases by sex and 5-year age group, and participation rates were high for both cases and controls[28], minimizing the potential for selection bias. Importantly, the demographic characteristics of the subpopulation with available CRS were similar to those of the overall study population (Supplementary Fig. 5), which in turn is representative of the general southern Chinese population[28], suggesting that the subpopulation with available CRS is informative about NPC risk in the general population. Hence, missing data were unlikely to result in a strong selection bias in our study, and our results appear to be generalizable to the broader population of southern China.

Given that human SNPs were successfully genotyped in 96.7% of saliva samples, EBV genotyping failure was mainly, if not solely, due to low EBV DNA concentrations. Because EBV is normally shed in saliva during the viral lytic cycle that occurs intermittently in the oral cavity, repeated saliva sampling may facilitate EBV DNA genotyping by capturing lytic EBV infection in individuals for whom initial sampling took place during latent EBV infection[3,29]. To investigate this possibility, we tested 19 repeated saliva samples and found that re-sampling during a four-week interval indeed increased the genotyping rate to 95% (Supplementary Table 8). These findings indicate that repeated collection of saliva samples can substantially increase the success rate of EBV genotyping, thereby maximizing the generalizability and the potential public health utility of the CRS.

In conclusion, we developed and validated a comprehensive risk prediction model for accurate NPC risk stratification using a CRS that combines EBV and human genetic variants, along with key epidemiological risk factors. Such a CRS could facilitate personalized risk prediction of NPC in southern China, and possibly in other endemic populations. The CRS substantially increased the accuracy of NPC screening when combined with the current standard EBV-serology-based approach to NPC screening. This study has important implications for the development of useful protocols to identify high-risk individuals for screening and early diagnosis of NPC. Future prospective cohort studies should further evaluate the CRS before it can be introduced in population screening and counseling programs.

## Methods

**Study population**. Study subjects were participants in a population-based NPC case-control study conducted in Guangdong and Guangxi Provinces, China, between 2010 and 2014. The study design has previously been described in detail[28,30]. Briefly, the patients with NPC and controls were enrolled from Guangdong and Guangxi Provinces between 2010 and 2014. The patients with NPC using the following eligibility criteria: (i) histological confirmation of NPC, (ii) age less than 80 years old, (iii) no treatment for NPC, and (iv) residence in Zhaoqing City, Guangdong Province and in Wuzhou City and Guiping and Pingnan Counties, Guangxi Province. Among the 1306 eligible patients with NPC who were recruited in Guangdong and 1248 recruited in Guangxi through rapid case ascertainment systems involving a network of local physicians, saliva DNA were available for 978 patients in Guangdong and 732 patients in Guangxi. Through random selection from total population registries, 1356 population-based control subjects in Guangdong and 1292 subjects in Guangxi with no history of malignancy were identified and enrolled with frequency matching to the cases by sex, 5-year age group, and areas of residence; saliva DNA were available for 1131 of the eligible Guangdong controls and 1115 of Guangxi controls. Each participant completed an in-person, structured interview conducted by a trained interviewer.

This study was approved by the Institutional Ethics Committee of the Sun Yat-sen University Cancer Center, Guangdong, China. Written informed consent was obtained from all participants.

**Sample (saliva) collection and DNA extraction**. Saliva samples were collected from 978 cases and 1131 controls in Guangdong and 732 cases and 1115 controls in Guangxi. Samples were mixed with an equal volume of prepared lysis buffer (50 mM Tris, pH 8.0, 50 mM EDTA, 50 mM sucrose, 100 mM NaCl, and 1% SDS) and stored at −80 °C. DNA was automatically extracted from 400 μl of saliva using Chemagic STAR (Hamilton Robotics, Sweden) according to the manufacturer's instructions.

**Genotyping of EBV and human genetic variants**. This study included the seven human SNPs (rs1412819, rs28421666, rs2860580, rs2894207, rs31489, rs6774494 and rs9510787) and three EBV SNPs (162215, 162476 and 163364) reported by previous GWASs and NPC-EBV studies. The reported summary statistics of these variants are shown in Supplementary Table 9.

Seven human SNPs (rs1412819, rs28421666, rs2860580, rs2894207, rs31489, rs6774494 and rs9510787) and three EBV SNPs (162215, 162476 and 163364) were genotyped in saliva DNA using the MassArray iPLEX (Agena Bioscience, California) (primers information in Supplementary Table 10). Our previous studies confirmed that the SNPs genotyped by the MassArray iPLEX and Sanger sequencing were 97.55% concordant[3]. A fixed position in the human albumin gene was used as a positive control. The genotyping completion rate for all seven human SNPs was 97%. Approximately half of the samples, 955 (55.8%) from cases and 1423 (63.4%) from controls, were successfully genotyped for all three EBV SNPs. Among all demographic characteristics (sex, age), epidemiological risk factors (family history of NPC, smoking, salted-fish consumption, educational level), and bio-sample characteristics (saliva EBV DNA copy number and VCA-/EBNA-1 IgA titer) considered, EBV DNA level ($P = 4.89E−125$) and smoking ($P = 6.95E−3$) were the only factor analyzed that was significantly associated with EBV genotyping failure in multivariate logistic regression. Because EBV is shed in saliva only during the viral lytic cycle that occurs periodically in normal (including healthy) oral epithelial cells[29,31–33], genotyping failure was mainly, if not solely, due to low EBV DNA concentration.

To assess potential selection bias, we evaluated whether missingness of EBV genotyping data was associated with case-control status or epidemiological risk factors for NPC. Based on our findings, which suggested limited bias (described in results), we excluded participants with missing EBV and host genotype data and missing covariate information from the analysis. Finally, 465 NPC cases and 589 controls from Guangdong province were included in the training dataset, and a non-overlapping set of 427 NPC cases and 751 controls from Guangxi province were included in the validation dataset. The training dataset in the current study has been included in the initial study describing the three EBV variants by Xu et al.[3], while the validation dataset of the current study is new and completely independent from the prior study.

**Comprehensive risk score**. The CRS was derived for each individual using the following formula:

$$CRS = \sum_{n=1\ldots k} \beta_n X_n. \tag{1}$$

where $X$ is the value of an individual risk factor and $\beta$ is the natural logarithm of the odds ratio (OR) for developing NPC for the specific risk factor. The effect size of each risk factor ($\beta$) was derived from a logistic regression analysis of the training dataset, except for family history, for which we adopted an estimate from literature review[5]. We built three risk prediction models using logistic regression: model 1, including only epidemiological risk factors (smoking status, salted-fish consumption, educational level, and family history of NPC); model 2, additionally including the two human SNPs (rs2860580 and rs2894207) that were significantly associated with NPC in the training dataset; and model 3 (the CRS model), including all factors above and additionally the three EBV SNPs. We performed a receiver operating characteristic (ROC) curve analysis and 10-fold cross-validation to assess the performance of the three models in discriminating NPC cases from controls. Net reclassification improvement (NRI) analysis was used for model comparison by quantifying how well a new model correctly reclassified the subjects (R package: PredictABEL[34]. The calibration of the CRS model was assessed using the Hosmer-Lemeshow goodness-of-fit[35]. Finally, we used logistic regression models to estimate ORs for NPC in relation to categories of the CRS.

**VCA- and EBNA1-IgA enzyme-linked immunosorbent assay (ELISA)**. Serum samples were stored at −80 °C before use. Serum VCA- and EBNA1-IgA antibody levels were measured using commercial ELISA kits (VCA-IgA, Euroimmun, Germany and EBNA1-IgA, Zhongshan Biotech, China) according to the manufacturers' instructions in the central laboratory of SYSUCC[14]. Levels of VCA- and EBNA1-IgA antibodies determined using ELISAs were assessed by performing a spectrophotometric measurement (Euroimmun Analyzer I, Germany) according to the manufacturers' instructions and standardized by calculating the ratio of the optical density (rOD) of the sample to the reference control provided with the ELISA kits. Among the samples with CRSs, serum samples were available from 401 patients with NPC and 479 controls for ELISA in the training dataset, and serum samples were available from 379 patients with NPC and 670 controls in the validation dataset.

**Evaluation of risk prediction by combining the CRS with EBV VCA-/EBNA1-IgA antibodies**. The serum VCA- and EBNA1-IgA antibody model has been reported and applied to identify high-risk groups from populations in endemic areas[13,14]. The serum antibody levels were used to calculate the probability of having NPC (Logit $P$) with the following logistic regression model:

$$\text{Logit } P = -3.934 + 2.203 \times \text{rOD\_VCA} + 4.797 \times \text{rOD\_EBNA1} \quad (2)$$

where rOD_VCA and rOD_EBNA1 are the serum antibody levels (rOD) of VCA- and EBNA1-IgA[12]. During screening, participants with $P \geq 0.98$ were identified as the "high-risk" group.

The EBV antibody model was combined with different CRS cutoff values. Subjects with an antibody model value of $P \geq 0.98$ and a higher CRS than the specific cutoff value were defined as the "high-risk" group. Next, we evaluated the specificity and sensitivity of the new combined models in training, validation and combined datasets. PPVs and NPVs were estimated by calculating the model specificity, sensitivity, and the prevalence of NPC.

$$\text{Prevalence} = (\text{Incidence rate}) \times (\text{Average Duration of Disease}) \quad (3)$$

$$\text{PPV} = \frac{\text{Sensitivity} \times \text{Prevalence}}{\text{Sensitivity} \times \text{Prevalence} + (1 - \text{Specificity}) \times (1 - \text{Prevalence})} \quad (4)$$

$$\text{NPV} = \frac{\text{Specificity} \times (1 - \text{Prevalence})}{(1 - \text{Sensitivity}) \times \text{Prevalence} + \text{Specificity} \times (1 - \text{Prevalence})} \quad (5)$$

The incidence rate of NPC in endemic areas of southern China was obtained from a recent report by Ji, M.F. et al[13]. The average duration of NPC was reported by the International Agency for Research on Cancer (IARC)[36].

Analyses were conducted in R version 3.6.1. All tests of statistical significance were two-sided.

**Reporting summary**. Further information on research design is available in the Nature Research Reporting Summary linked to this article.

## Data availability
The genotyping data and sample covariate information reported in this paper have been deposited in the European Nucleotide Archive (ENA) at the EMBL-EBI under accession number PRJEB46183 (accessible at https://www.ebi.ac.uk/ena/browser/view/PRJEB46183) and National Genomics Data Center (NGDC, Nucleic Acids Res 2021), Beijing Institute of Genomics, Chinese Academy of Sciences, under accession number PRJCA005360 (accessible at https://ngdc.cncb.ac.cn/bioproject/browse/PRJCA005360). Source data are provided with this paper.

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

## Acknowledgements

This work was supported by grants from the National Key Research and Development Program of China (grant number 2020YFC1316900), the National Natural Science Foundation of China (grant number 81872228 to M.X.;81430059 to Y.X.Z.), the Guangdong Natural Science Fund for Distinguished Young Scholars (grant number 2020B1515020002 to M.X.), the Fundamental Research Funds for the Central Universities to M.X., and the National Cancer Institute at the US National Institutes of Health (grant number R01CA115873-01 to H.A. and Y.X.Z.). We thank all the participants for their generous support of the current study. We also thank Dr. Zhiwei Liu, Dr. Fang Fang and Dr. Weiwei Zhai for helpful discussions on statistical analyses and the manuscript.

## Author contributions

M.X., Y.-X.Z. and Z.Z. conceived and designed the study. Xiang Zhou, Xiao Zhang, S.Z., G.-F.F., Yufeng Chen, Q.-S.F. and Yijun Chen performed the sample preparation, quality control and genotyping. M.X., Xiang Zhou, Z.L. and J.L. conducted the statistical analyses. W.Y., H.-O.A., E.T.C., S.-M.C., Y.-L.C. and Z.Z. participated in the case-control study design, subject recruitment, and sample collection. The manuscript was drafted by M.X. and Xiang Zhou.

## Competing interests

The authors declare no competing interests.
