## [Peer Review File · Nature Communications]

A Comprehensive Risk Score for Effective Risk Stratification and Screening of Nasopharyngeal CarcinomaReviewers' comments:

Reviewer #1 (Remarks to the Author):

An interesting article that provides impressive results in risk discrimination of NPC. The training and validation design appears appropriate and methods appropriate. The three EBV variants appear particularly striking. Are the two cohorts design independent to studies used in the initial description of these variants?

The majority of the predictive value appears to stem from the three EBV variants (supplementary figure 1 for example). In the context, the assay was only partial successful (~50% of individuals due to the complexities of assaying the EBV genome), consequence on the model (differed by validation / optimization cohort), how was missed data modelled? Are these variants correlated? How was this correlation modelled in the risk prediction?

Is there any difference by tumour stage (particularly table 2).

Reviewer #2 (Remarks to the Author):

In this report, the authors have established combined NPC risk scores that induces multiple epidemiology factors, 3 EBV SNPs of BALF2 gene and 7 human germline SNPs. They suggested that combining the top 20% CRS strata with EBV serology as new approach for screening high-risk individuals in Southern China, an endemic area of NPC. The inclusion of CRS has high improved the PPV of EBV serology test from 4.7% to 43.5% for NPC screening. Instead of EBV serology, plasma EBV DNA test is a well-known cancer markers of NPC and has a high positive value of 11.0% to 19.6% for NPC screening in Southern Chinese (Chan et al. N Engl J Med, 2017; Lam et al. Proc Natl Acad Sci U S A. 2018). The recent sequencing analysis of plasma EBV DNA SNPs further demonstrated its usefulness on predicating NPC risk (Lam et al., Clin Chem, 2020). Nevertheless, the authors have not included or discussed this high sensitivity and specificity test in this paper. Another major concern of the application of CRS for NPC screening is that only 50-60% of individual have detectable saliva EBV DNA for genotyping, the low detection rate may limit the usefulness of CRS in population screening. The relationship of EBV stains in the persistent memory B cells, activated replication lymphocytes, NPC tumors and saliva in NPC patients and infected individuals is still needed to clarify.

Major comments:

- In addition to the 3 SNPs of BALF2, the authors should explore whether the inclusion of the multiple NPC-related EBV SNPs, eg. EBER SNPs in Southern Chinese in the EBV genetic model will improve the prediction of NPC risk.
- The author should clarify whether saliva DNA is a representative source of EBV genotype for NPC screening. It was previously reported multiple EBV stains were found in the saliva samples. The CRS for multiple EBV stains should be clarified. Furthermore, I also concern whether distinct EBV stains are detected in the tumor, peripheral blood and saliva samples of same NPC patient. Are there any data from longitudinal studies to show there is persistent identical EBV stain in the saliva sample in healthy individuals?
- Except for family history, the epidemiological factors seem to have limited impacts for NPC risk prediction. As shown in Supplementary Figure 1, the ROC curves of EBV genetic and Comprehensive models are overlapped, suggesting there are no advantage to include epidemiological factor, even Host genetic factors, for predicating NPC risk. In Supplementary Table 3, the performance of EBV genetic models is identical to that of comprehensive model in the validation dataset. No obvious advantage of comprehensive model is noted.
- In Supplementary Table 4, the authors should determine the Net reclassification index of "3 EBV SNPs" vs "3 EBV SNPs + 7 Host SNPs" vs "3EBV SNPs + & Host SNPs + Epidemiology". The findings may show the limited advantages for adding epidemiology (or host SNPs) in the score system.
- The findings for risk assessment and predication values of CRS may be bias since only the individuals with detectable saliva EBV DNA were included in their study. As shown in the methods section, the EBV genotyping can be determined in only 50-60% of patients and controls. This

indicates that CRS is applicable to predict NPC risk in only half of the Southern Chinese population. Furthermore, the EBV DNA in saliva is believed to be influenced by the host immunological status and viral replication. Multiple saliva DNA assay for the patients/controls should be conducted to increase the successful rate of EBV genotyping. Nevertheless, the authors should also determine whether the presence or amount of saliva EBV copies may also be associated with the NPC risk.

- For the combined CRS and EBV serology for NPC screening, the author concluded that the PPV is increased from 4.70% to 43.47%. However, since half of the cases may have undetectable saliva EBV DNA, CRS should not be available in these cases. Thus, the PPV and NPV seem to be derived from this selected population, but not for the general population. It is likely that CRS may not be available in most of the EBV serology negative cases.

- The host and EBV SNPs associated with NPC were reported from the studies in Southern China. Are the high risk EBV SNPs also found in the NPC patients from other endemic regions in Southeast Asia such as Malaysia, Singapore and Indonesia according to the available EBV genome data?

- Several supplementary data should be included in the main text, e.g. Supplementary figure 1, Table 3 and Table 4.

Reviewer #3 (Remarks to the Author):

"Comprehensive Risk Scores Based on Epstein-Barr Virus and Genetic Variants for Effective Nasopharyngeal Carcinoma Screening",

This paper discusses risk prediction for NPC, a rare cancer, by combining EBV and human genetic information with personal characteristics (sex, age, smoking, etc) into comprehensive risk scores (CRSs) and study the impact of addition of EBV antibody data, based on data from a large NPC case-control study conducted in two locations in China.

Main points:

Controls were frequency matched to cases on sex and age, thus gender and age effects could not be estimated. The matching should be mentioned in the main paper as it is a key feature of the study. Because of this matching, the association parameters (odds ratios) for sex and age relevant for the general population cannot be estimated. The estimates that are obtained here are only reflecting the study design. Therefore using these estimates in a risk prediction model is incorrect (line 103). Assessing performance of a model that contains age and sex parameters estimated from these data is thus not appropriate.

As the training set was used to obtain estimates of association for the studied predictors, it should not be used in any part of the paper to evaluate model performance, that should be strictly restricted to the validation (test) set. These evaluations need to be removed (some, but not all occurrences of using the training data for validation are mentioned below)

Other detailed comments:

- From a practical clinical and public health application perspective: how easy and expensive is it to obtain the EBV and human SNPs? Does EBV viral infection ever clear? How reliably can viral genetic information be measured?
- Line 106: do not show AUC for training data, it is biased
- Figure 3: remove training set from figure
- Line 100: if the 5 human SNPs are not significant, why include them in the risk model?
- Lines 120-121: how was "did not contribute significantly to accuracy of risk" evaluated/quantified?
- Line 133: to talk about event rates is misleading: Figure 1B in Supplement does not assess prediction in the population, as only the proportions of cases in a case control study in deciles of the score are evaluated, i.e. the Hosmer Lemeshow test is assessing if the relative risks in deciles agree between the training and validation population. This is only assessing the goodness of fit of the relative risk model and is not the same as saying something about the actual fit of an absolute risk model.

- Line 154: estimated risks in the training set need to be removed.
- Line 158: the datasets should not be combined.
- Table 2: please compute all the performance characteristics using only the validation data, not the combined dataset
- Line 219: if 20% of the population at highest NPC risk need to be screened further, that is a large number. How many cases will be detected in those 20% at highest risk based on the CRS prediction?
- I do not understand the computation of the absolute risk in the supplement, please clarify

We thank the editor and reviewers for thoroughly examining our manuscript and for providing valuable comments and suggestions. The points raised have helped us to improve the presentation of our data and the manuscript overall. Below, we have provided our responses to all points raised by the three reviewers and have indicated the changes made to the revised manuscript.

Reviewers' comments:

Reviewer #1 (Remarks to the Author):

An interesting article that provides impressive results in risk discrimination of NPC. The training and validation design appears appropriate and methods appropriate. The three EBV variants appear particularly striking. Are the two cohorts design independent to studies used in the initial description of these variants?

Response: We thank the reviewer for the positive comments. The initial study describing the three EBV variants published in Nature Genetics (our prior study) had two phases—a discovery phase and a validation phase. In the current study, we also have training and validation phases, which are both independent from the discovery phase of the prior study. The two studies have one overlapping dataset that was used in the validation phase of prior study and the training phase of the current study; however, the validation dataset of the current study is new and completely independent from the prior study. We have added this description of the datasets used in these two studies in the revised manuscript (Lines 359-364).

The majority of the predictive value appears to stem from the three EBV variants (supplementary figure 1 for example). In the context, the assay was only partial successful (~50% of individuals due to the complexities of assaying the EBV genome), consequence on the model (differed by validation / optimization cohort), how was missed data modelled? Are these variants correlated? How was this correlation modelled in the risk prediction?

Response: The reviewer's understanding is correct that the three EBV variants are the predominant contributors to the risk prediction performance. After we included the three EBV variants in the predictive model, the area under the curve (AUC) was increased from

0.62 to 0.77 in the validation dataset, compared with the model including only epidemiological and human genetic factors (**Figure 2A** and **Table 2**).

To address the reviewer's question of whether the incomplete EBV genotyping rate introduced any bias into the study, we have added a comparison between the subjects with EBV genotyping data and those without. Genotyping information on human SNPs was obtained for 96.7% (3825/3955) of the participants, whereas EBV variant information on all three positions was available for 892 cases (52.1%) and 1340 controls (59.7%). The validation dataset had a higher genotyping rate than the training dataset; this appeared to be explained by saliva EBV DNA quantity, since the EBV genotyping rate did not differ statistically significantly between the two groups after adjustment for EBV DNA level ($P=0.17$). Most importantly, the good performance of the CRS in both the training and the validation datasets indicates that bias was not introduced by differences in the EBV genotyping rate between the two groups.

The success rate for EBV genotyping is likely dictated by the quantity of EBV DNA in saliva. As shown in the literature, periodic lytic EBV production in the oral epithelium is hypothesized to be the main source of the fluctuation observed for saliva EBV DNA¹. Because lytic EBV production in oral cavity happens intermittently under normal conditions in healthy donors, we anticipate that the fluctuation of saliva EBV DNA level, and therefore, the pattern of missing EBV genotyping data is approximately random. Indeed, in our dataset, the missingness of EBV genotyping data did not differ by most covariates analyzed, including age, salted-fish consumption, educational level and family history of NPC (**Supplementary Figure 1A**), as well as EBV serology (see **Appended Figure 4B** on page 13, in response to the fifth question raised by reviewer 2). A lower frequency of missing EBV genotype data was observed among smokers than non-smokers, and among men (who are substantially more likely to be smokers) than women (**Supplementary Figure 1B**). These findings are concordant with reports that smoking stimulates EBV lytic production^{2,3}, which increases the chance of EBV being genotyped. However, because the association between smoking and NPC risk is modest, with relative risk of only 1.1-1.5, any bias caused by smoking in our dataset would be small. Additionally, smoking status and educational level (as an indicator of socioeconomic status) were included as covariates in our models to control for confounding. (Age and sex were not included in the CRS model due to the frequency matched design.) To

avoid introducing bias caused by imputation errors, we excluded subjects with missing EBV genotypes.

Taken together, based on the lack of correlation between missingness and most covariates analyzed, and the weak effect of smoking on NPC, the case-control subset included in the present study appears to be representative of the overall case-control population, which in turn had high participation rates and was representative of the underlying source population; therefore, the incomplete EBV genotyping data should not have an appreciable effect on the internal validity of our conclusions. We have added the analyses and discussion of the distribution of missing data to the results (Lines 128-134, **Supplementary Figure 1**) and discussion (Lines 291-301). While acknowledging this limitation of missing data, we have also added a discussion that repeated saliva sampling and the adoption of new technologies, such as digital PCR, may improve the genotyping rate, thereby improving the external validity, in future screening studies (Lines 302-310).

The three EBV variants were correlated with each other. The correlation R^2 is shown in **Appended Table 1** below. All three variants were included simultaneously in the same prediction model, such that they were mutually adjusted and also adjusted for other covariates, without excessive collinearity leading to inflated errors. Thus, even after accounting for their intercorrelation, all three variants were independently associated with NPC risk in the validation dataset.

Appended Table 1. Correlation R^2 between three EBV SNPs in the study subjects.

Variables of interest	EBV162476_C	EBV163364_T
EBV162215_C	0.787	0.561
EBV162476_C	-----	0.688
EBV163364_T		-----

Is there any difference by tumour stage (particularly table 2).

Response: We found no difference by tumor stages between the high-risk and low-risk groups in the combined dataset. The NPC cases identified by the CRS and serology combined test in Table 2 contained both early- and late-stage cases, and the stage distribution did not differ from that among the unidentified cases (**Appended Figure 1**).

Appended Figure 1

Appended Figure 1. Tumor stages in high-risk and low-risk groups. The P value was calculated using χ^2 test.

Reviewer #2 (Remarks to the Author):

In this report, the authors have established combined NPC risk scores that induces multiple epidemiology factors, 3 EBV SNPs of BALF2 gene and 7 human germline SNPs. They suggested that combining the top 20% CRS strata with EBV serology as new approach for screening high-risk individuals in Southern China, an endemic area of NPC. The inclusion of CRS has high improved the PPV of EBV serology test from 4.7% to 43.5% for NPC screening. Instead of EBV serology, plasma EBV DNA test is a well-known cancer markers of NPC and has a high positive value of 11.0% to 19.6% for NPC screening in Southern Chinese (Chan et al. N Engl J Med, 2017; Lam et al. Proc Natl Acad Sci U S A. 2018). The recent sequencing analysis of plasma EBV DNA SNPs further demonstrated its usefulness on predicating NPC risk (Lam et al., Clin Chem, 2020). Nevertheless, the authors have not included or discussed this high sensitivity and specificity test in this paper. Another major concern of the application of CRS for NPC screening is that only 50-60% of individual have detectable saliva EBV DNA for genotyping, the low detection rate may limit the usefulness of CRS in population screening. The relationship of EBV stains in the persistent memory B cells, activated replication lymphocytes, NPC tumors and saliva in NPC patients and infected individuals is still needed to clarify.

Response: We appreciate the reviewer's constructive critiques. For the benefit of the editor(s) and other reviewers, here we would like to place this reviewer's comments in context by providing some brief background. Because NPC is an endemic cancer (with a prevalence similar to that of lung cancer in southern China), many groups have sought to improve public health by developing NPC screening programs. Screening tests have been based on two technologies, plasma EBV DNA and EBV serology (serum EBV IgA antibodies). We agree with the reviewer that plasma EBV DNA, pioneered by Prof. Dennis Lo and colleagues, is a well-known and high-performance marker used as an indicator of NPC prognosis and early diagnosis^{4,5}. The value of plasma EBV DNA for early NPC diagnosis has been well demonstrated in the Chinese male population from Hong Kong. The other NPC screening strategy, based on EBV serology, has been conducted by colleagues from mainland China, Taiwan and southeast Asia, and has been shown to be quite powerful for NPC risk prediction^{6,7}.

These two methods appear to have different value in terms of NPC screening, early diagnosis and prognosis. Prof. Dennis Lo and his colleagues have also shown that plasma EBV DNA is derived largely from apoptotic cancer cells. Hence, plasma EBV DNA is anticipated to be more useful for NPC early diagnosis and prognosis than for risk prediction and NPC screening. In contrast, serum EBV EBNA-1- and VCA-IgA antibodies have been shown to be elevated 3-5 years before NPC tumors can be detected, with very high specificity⁷. For this reason, EBV serology markers are currently anticipated to be more effective for NPC screening, and have been widely used for this purpose in mainland China and Taiwan^{6,7}, although the issue is not settled and this field remains an area of active and fruitful research.

There has also been a common limitation, low positive predictive value (PPV), for both of these methods when used for NPC screening. So far, screening studies based on plasma EBV DNA have been shown to be successful in a subset of the at-risk population, specifically, 40- to 62-year-old males, among whom NPC risk peaks⁸. Because NPC incidence among women is about 1/3 of the incidence among men, if this method is used for population screening of men and women across all adult age groups, the PPV for screening may be substantially lower in the general population. Meanwhile, for EBV serology tests, the PPV for NPC screening in all adult age groups and in both genders combined has been reported to be 1-

5%^{6,9}. Hence, there still is a strong need for a more accurate screening strategy in NPC-endemic populations.

By explaining about 80% of the overall risk for NPC in southern China, the recently identified NPC-associated EBV risk subtypes provide promising biomarkers for advancing population screening in this region (Xu et al., *Nature Genetics* 2019). The recent study mentioned by this reviewer (Lam et al., *Clinical Chemistry* 2020), although relatively small in size, further suggests that the combination of EBV risk variants and plasma EBV DNA load information may improve the PPV based on plasma EBV DNA alone¹⁰. Presented in the current work, we further demonstrated with the first large, strictly population-based NPC case-control study in southern China that high-risk EBV variants together with EBV serology offer a useful basis for NPC screening.

Reviewer 2 suggested including the plasma EBV DNA test in our analysis and compare our method (based on EBV subtypes and serology tests) against plasma EBV DNA. We agree with the reviewer that this comparison would be informative, and in the future, it may be important to evaluate whether one approach is superior (although both strategies may co-exist, based on the availability of appropriate technology and/or the target population). However, logistically it is challenging and extremely time-intensive to establish a new cohort of healthy individuals with bio-samples (blood, saliva and nasal swab) collected simultaneously, and to follow such a cohort forward over several years to identify new NPC cases. Moreover, because EBV DNA is present only at trace levels that are much lower in blood than in saliva in healthy populations, it can be difficult to absolutely quantify plasma EBV DNA. A well-established, standardized protocol to evaluate absolute plasma EBV DNA is not yet widely available. Due to these substantial logistical constraints, it is not feasible at present for us to carry out this comparison. We hope that a collaborative epidemiological study with colleagues who have expertise in plasma EBV testing will take the field forward in the near future.

To directly address the reviewer's suggestion, we have now added a deeper discussion of these issues, especially highlighting the accuracy and the application of plasma EBV DNA test (Lines 240-243), as well as the value of future comparisons between plasma EBV DNA and our newly developed approach.

Major comments:

- In addition to the 3 SNPs of BALF2, the authors should explore whether the inclusion of the multiple NPC-related EBV SNPs, eg. EBER SNPs in Southern Chinese in the EBV genetic model will improve the prediction of NPC risk.

Response: We thank the reviewer for this question. The EBER SNPs are in almost completely linkage disequilibrium with SNP 163364, one of the three BALF2 SNPs included in EBV strains from southern Chinese. The correlation R^2 between EBER SNP 7048 and SNP 163364 was 0.82 in our training dataset. Importantly, in our original discovery study (Xu et al., *Nature Genetics* 2019), conditioning on the three BALF2 SNPs abolished the effects of other EBV variants on NPC. Therefore, including EBER SNPs would have limited impact on the prediction of NPC risk in the southern Chinese population, in which both our prior study and the current study are based.

- The author should clarify whether saliva DNA is a representative source of EBV genotype for NPC screening. It was previously reported multiple EBV stains were found in the saliva samples. The CRS for multiple EBV stains should be clarified. Furthermore, I also concern whether distinct EBV stains are detected in the tumor, peripheral blood and saliva samples of same NPC patient. Are there any data from longitudinal studies to show there is persistent identical EBV stain in the saliva sample in healthy individuals?

Response: In our earlier study (Xu et al, *Nature Genetics* 2019), using EBV whole-genome sequencing from one paired sample, we provided some evidence that the EBV strain in saliva and tumor tissue were the same. In the revised version of the present manuscript, we have now added saliva-tissue paired sample genotyping for 20 NPC cases, and showed that the same EBV variants in NPC tumors were detected in the saliva DNA in at least 90% of the pairs tested (**Appended Table 2**). In another previous paper (Cui et al, *Oncotarget* 2017), we have showed that the consistency between a single EBV variant 155391 detected from tumor and saliva was 98% (49/50), and EBV DNA was too rare to be genotyped 9.71% (10/103) in blood from most healthy individuals¹¹. By combining these two lines of evidence, our data show that saliva DNA is valid and a more suitable source than blood for analyzing EBV variants.

Appended Table 2. Concordance rate of EBV variant genotypes in paired saliva and tumor samples.

		Tumor		Concordance rate*
		High risk	Low risk	
EBV162215				
	High risk	17 ^a	0 ^b	100%
	Low risk	0 ^c	3 ^d	
EBV162476				
Saliva	High risk	16 ^a	0 ^b	100%
	Low risk	0 ^c	4 ^c	
EBV163364				
	High risk	13 ^a	1 ^b	90%
	Low risk	1 ^c	5 ^c	

* Concordance rate calculated by (a+d)/(a+b+c+d).

Regarding to the comment on single-strain *versus* multi-strain EBV infection, we have now tested whether including multiple infection (observed in 4.9% of cases and 7.3% of controls) would affect the prediction performance. Our results show that the risk strain had similar risk effects on NPC, and the prediction performance remained almost the same in the training and validation datasets, regardless of whether we included multiple infection in the prediction model (**Appended Table 3**). Because multiple EBV infection did not significantly contribute to the prediction model, we have not included it in the final comprehensive model.

Appended Table 3. Performance of models including multiple EBV infection or not for distinguishing NPC cases from controls.

Model	Training dataset			Validation dataset		
	AUC	95%CI	P value	AUC	95%CI	P value
Comprehensive model*	0.778	0.75-0.806	3.92E-01	0.779	0.752-0.806	9.82E-02
Comprehensive model + multiple infection†	0.780	0.752-0.807	-----	0.783	0.756-0.809	-----

*The comprehensive model included epidemiological risk factors (smoking, salted fish consumption, education and family history of NPC), 2 host SNPs (rs2860580 and rs2894207) and 3 EBV SNPs (EBV162215, EBV162476 and EBV163364)

†This model included all covariates in the comprehensive model, as well as infection with multiple EBV strains as an additional covariate

The reviewer raises an interesting and important question of whether there are persistently identical EBV strains in serially collected saliva samples from healthy individuals. As far as we know, this question remains to be studied for the field.

- Except for family history, the epidemiological factors seem to have limited impacts for NPC risk prediction. As shown in Supplementary Figure 1, the ROC curves of EBV genetic and Comprehensive models are overlapped, suggesting there are no advantage to include epidemiological factor, even Host genetic factors, for predicating NPC risk. In Supplementary Table 3, the performance of EBV genetic models is identical to that of comprehensive model in the validation dataset. No obvious advantage of comprehensive model is noted.

Response: The reviewer is quite right that the epidemiological (i.e., behavioral and environmental) factors and human genetic factors have a limited contribution to the risk prediction model in terms of absolute magnitude, after accounting for the three EBV genetic variants. After considering numerous known and suspected epidemiological risk factors for NPC for which we collected data, e.g., a family history of NPC, salted-fish consumption, smoking, socioeconomic status, educational level, tea drinking, wood dust exposure, formaldehyde exposure and oral hygiene, we chose to include four factors: a family history of NPC, salted-fish consumption, smoking, and educational level. These are well-defined risk factors that have been consistently associated with NPC risk in epidemiological studies and are relatively straightforward to recall and quantify. These epidemiological factors collectively had a statistically significant, albeit small, contribution to prediction performance, increasing the AUC by about 2% in the validation dataset ($P = 2.87E-3$), compared with the model including only EBV variants (**Supplementary Table 3**). Including smoking status and educational level (as an indicator of socioeconomic status) in the logistic regression model also helped to control for potential confounding effects (for instance, by factors related to availability of EBV DNA in saliva). The model calibration was also improved when the four epidemiological factors and two host HLA SNPs were included, as demonstrated by increasing R^2 in Hosmer-Lemeshow analyses (**Appended Figure 2**). Hence, we included both the epidemiological factors and the host genetic factors in the comprehensive model.

Appended Figure 2

Appended Figure 2. Hosmer-Lemeshow goodness-of-fit analyses of the three risk prediction models in the training and validation datasets. Diagonal, the line of perfect fit between observed and predicted probability. Blackspot, observed probabilities for 10 groups of equal size. R², Nagelkerke pseudo-R². P, Pearson chi-squared goodness-of-fit test.

- In Supplementary Table 4, the authors should determine the Net reclassification index of “3 EBV SNPs” vs “3 EBV SNPs + 7 Host SNPs” vs “3EBV SNPs + & Host SNPs + Epidemiology”. The findings may show the limited advantages for adding epidemiology (or host SNPs) in the score system.

Response: We agree that the contributions of the epidemiological factors and host genetic factors to the risk prediction model were limited in magnitude. As suggested by the reviewer, we have now added the Net reclassification index (NRI) and ROC analyses to compare the three models, “3 EBV SNPs” vs “3 EBV SNPs + 2 Host SNPs” vs “3EBV SNPs + 2 Host SNPs + Epidemiology”. Even though their absolute effect sizes were relatively small, the two HLA SNPs and the four epidemiological factors made statistically significant contributions to the model in the validation dataset, with AUC increased by 1.6% ($P = 1.20E-02$) and 1.9% ($P = 2.87E-03$) and NRI increased by 21% (95% CI : 15%-27%) and 9% (95% CI: 5%-12%) for HLA SNPs and epidemiological factors, respectively (**Supplementary Table 3 and Supplementary Table 4**). As discussed in response to the reviewer’s prior comment, the model calibration was also improved when epidemiological factors and two HLA SNPs were included, as demonstrated by Hosmer-Lemeshow analyses (**Appended Figure 2**). Finally, from a practical standpoint, epidemiological data are relatively easy and inexpensive to

collect if bio-samples are already being obtained, so adding these factors would not make the risk prediction model more difficult to implement in the real world. Therefore, we included these two human SNPs and epidemiological factors in the comprehensive model.

- The findings for risk assessment and predication values of CRS may be bias since only the individuals with detectable saliva EBV DNA were included in their study. As shown in the methods section, the EBV genotyping can be determined in only 50-60% of patents and controls. This indicates that CRS is applicable to predict NPC risk in only half of the Southern Chinese population. Furthermore, the EBV DNA in saliva is believed to be influenced by the host immunological status and viral replication. Multiple saliva DNA assay for the patients/controls should be conduct to increase the successful rate of EBV genotyping. Nevertheless, the authors should also determine whether the presence or amount of saliva EBV copies may also be associated with the NPC risk.

Response: We appreciate the reviewer's comments. As discussed in our response to the second comment from reviewer 1, we agree that currently the CRS is applicable for NPC risk prediction only in those having EBV genotype information. In the current training and validation datasets, we found that the presence (~52% of cases and ~60% of controls) and amount of saliva EBV DNA (**Appended Figure 3**) were not associated with increased risk of NPC, consistent with a recent report¹². As we noted earlier, because periodic lytic EBV production occurs normally in healthy individuals, the availability of EBV DNA should be approximately random. In addition, the vast majority of adults should shed EBV DNA in saliva at some time, making them eligible for NPC risk prediction using the CRS.

As described in our response to the second comment from reviewer 1, we further investigated the pattern of missing EBV genotyping data to evaluate the potential for bias. Taken together, based on the lack of correlation between missingness and most covariates analyzed, and the weak effect of smoking on NPC, the case-control subset included in the present study appears to be representative of the overall case-control population, which in turn had high participation rates and was representative of the underlying source population; therefore, the incomplete EBV genotyping data should not have an appreciable effect on the internal validity of our conclusions.

In accordance with this reviewer's suggestion, we have modified our manuscript to include a more conservative statement that the CRS currently is applicable only to individuals in the southern Chinese population with EBV genotype information available; this statement has been added in the introduction (Lines 111-114), results (Lines 124-126) and discussion (Lines 291-292). We have also added the results and discussion of our analyses of missing data (Lines 128-135 and 291-301).

Appended Figure 3

Appended Figure 3. Distribution of EBV DNA abundance in saliva samples from 1674 patients with NPC and 2202 controls. The 95% confidence intervals of the mean values are shown as black bars. The *P* value was calculated using a two-sided t-test.

We agree with the reviewer that the amount of EBV DNA in saliva may be affected by the EBV-specific immune response. However, so far, substantive evidence is lacking to identify the specific immune response to support this hypothesis, because the presence and level of EBV DNA in saliva only weakly correlated with serum VCA-IgA and EBNA1-IgA antibody levels in our datasets ($R < 0.1$, **Appended Figure 4A**). Even in the EBV-IgA-serology-negative (low-risk) controls without NPC, we found that the detection rate of saliva EBV DNA was very similar to that in EBV-IgA-serology-high-risk and -medium-risk controls (**Appended Figure 4B**). In line with the publication from Hadinoto et al¹, our results showing an increased genotyping success rate from repeated saliva sampling suggest that intermittent EBV replication in the oral cavity is the main source of the fluctuation observed for saliva EBV DNA. We have added the results and discussion on saliva EBV DNA and serology accordingly (Lines 126-128 and 302-310).

Appended Figure 4

Appended Figure 4. (A) Correlation between serum EBNA1-IgA/VCA-IgA antibody levels and EBV DNA level in healthy subjects. (B) Stacked bar plot representation of the distribution of EBV serum antibody (EBNA1-IgA/VCA-IgA) levels in healthy subjects.

The reviewer is also correct that multiple saliva DNA assays over time can increase the success rate of EBV genotyping. Taking this reviewer's suggestion, we conducted a pilot study showing that repeated saliva sampling within 4 weeks boosted the success rate of EBV genotyping to 95% (**Supplementary Table 6**). Although it is no longer possible to collect repeated bio-samples from the population-based NPC cases and controls in this study, the high EBV genotyping success rate with only two to three rounds of sampling supports the feasibility and utility of the CRS approach in general. We have added the discussion and results on multiple saliva sampling in the revised manuscript (Lines 302-310).

In summary, following this reviewer's comments and suggestions, we have revised our manuscript as follows: we have clarified that the current CRS is applicable to those with EBV genotype information available, and further showed that having missing EBV genotype data was not associated with the majority of known NPC risk factors, and is thus unlikely to cause strong bias in the current study. Finally, we investigated the reviewer's suggestion on

multiple saliva sampling and found that it can substantially increase saliva EBV genotyping rate after a fairly short time interval of four weeks.

- For the combine CRS and EBV serology for NPC screening, the author concluded that the PPV is increased from 4.70% to 43.47%. However, since half of the cases may have undetectable saliva EBV DNA, CRS should not be available in these cases. Thus, the PPV and NPV seem to be derived from this selected population, but not for the general population. It is likely that CRS may not be available in most of the EBV serology negative cases.

Response: As discussed in response to the reviewer's preceding comment, we agree with the reviewer that the current CRS model is applicable only to the subpopulation with EBV genotyping information. However, we found that missingness of EBV genotyping data was distributed randomly with respect to the majority of NPC risk factors, making it unlikely to be an appreciable source of bias (**Supplementary Figure 1**). Importantly, the demographic characteristics of the subpopulation with available CRS remain very similar to those of the complete study population, as well as the underlying general population in the study area (**Supplementary Figure 6**). Therefore, it is reasonable to estimate the PPV and NPV of the CRS for the general population of southern China based on study participants with available data.

We also agree with the reviewer that with multiple saliva sampling or improvement in genotyping technologies such as digital PCR, we may overcome this limitation in our future studies. Indeed, we are actively investigating both of these promising avenues of research. In the current study, we have added the results comparing the demographic characteristics between the subpopulation with available CRS and the complete study population (**Supplementary Figure 6**), and we have added a discussion of the bias potentially caused by missing data (Lines 128-134 and 291-301), as well as the potential of multiple sampling to improve the population coverage and external validity of CRS (Lines 302-310).

- The host and EBV SNPs associated with NPC was reported from the studies in Southern China. Are the high risk EBV SNPs were also found in the NPC patients from other endemic regions in Southeast Asia such as Malaysia, Singapore and Indonesia according to the available EBV genome data?

Response: The EBV subtypes from other populations are not yet known for us. Further international collaboration will be needed to address this question in the coming future.

- Several supplementary data should be included in the main text, e.g. Supplementary figure 1, Table 3 and Table 4.

Response: We thank the reviewer for this suggestion. We have added Figures 2 and Table 2 describing AUC and model comparison into the main text in the updated version.

Reviewer #3 (Remarks to the Author):

"Comprehensive Risk Scores Based on Epstein-Barr Virus and Genetic Variants for Effective Nasopharyngeal Carcinoma Screening",

This paper discusses risk prediction for NPC, a rare cancer, by combining EBV and human genetic information with personal characteristics (sex, age, smoking, etc) into comprehensive risk scores (CRSs) and study the impact of addition of EBV antibody data, based on data from a large NPC case-control study conducted in two locations in China.

Main points:

Controls were frequency matched to cases on sex and age, thus gender and age effects could not be estimated. The matching should be mentioned in the main paper as it is a key feature of the study. Because of this matching, the association parameters (odds ratios) for sex and age relevant for the general population cannot be estimated. The estimates that are obtained here are only reflecting the study design. Therefore using these estimates in a risk prediction model is incorrect (lines 103). Assessing performance of a model that contains age and sex parameters estimated from these data is thus not appropriate.

Response: We highly appreciate the reviewer's constructive comments and suggestions regarding the frequency matching design. We agree that the association parameters for sex and age for the general population cannot be estimated under the current matched design. In the revised version of our manuscript, we reanalyzed our data and built the CRS without

the matching covariates, sex and age, included in the logistic regression model (**Table 2 and Supplementary Table 2**). We also estimated absolute risks among men and women separately (**Figure 4 and Supplementary Figure 4**). The CRS maintained good performance for NPC risk stratification in the new analyses. We have revised our manuscript with these new results, and we have accordingly emphasized the matched design in the results (Lines 140-142), methods (Lines 325-329) and discussion (Lines 256-258).

As the training set was used to obtain estimates of association for the studied predictors, it should not be used in any part of the paper to evaluate model performance, that should be strictly restricted to the validation (test) set. These evaluations need to be removed (some, but not all occurrences of using the training data for validation are mentioned below)

Response: We thank the reviewer for this suggestion. In this revised version, we have moved the results for the training dataset from the main text to the supplementary materials, and omitted the results for the combined dataset. Only the results for the validation dataset are now included in the main text.

Other detailed comments:

- From a practical clinical and public health application perspective: how easy and expensive is it to obtain the EBV and human SNPs? Does EBV viral infection ever clear? How reliably can viral genetic information be measured?

Response: We thank the reviewer for the question from a practical and public health perspective. The current cost of human and EBV genotyping is approximately \$15.8 USD (saliva collection \$2.8 + DNA extraction \$3.6 + SNPs genotyping \$9.4 = \$15.8; or €12.9 EUR) in mainland China and much less expensive in Hong Kong and Singapore. We have included a discussion of cost and screening steps in the revised manuscript (Lines 270-284).

So far, the evidence suggests that EBV infection cannot be cleared, but instead establishes a lifelong persistent infection in memory B cells¹³. In healthy individuals, only 1 in 10⁵ to 10⁶ plasma cells carry latent EBV, so that it is very difficult to detect EBV in blood¹¹. However, EBV often re-infects the oral epithelial cells and undergoes a periodic lytic cycle, during which it sheds through saliva¹. Hence, in line with the results from the EBV genotyping

experiments on saliva and blood DNA (Cui et al, *Oncotarget* 2017), saliva is indeed a better source to genotype EBV than blood.

The Agena MassArray is a mature and very sensitive platform for viral and host genotyping with medium throughput. We have demonstrated the reliability of this method to genotype EBV variants, with high consistency (97.55%) compared with the conventional PCR-sanger sequencing method, in our previous publication (Xu et al, *Nature Genetics* 2019). This genotyping platform has been widely used for genotyping variants in humans and infectious agents. For example, during the pandemic of COVID 19, this platform has been widely applied for SARS-CoV-2 detection in the US and Europe.

- Line 106: do not show AUC for training data, it is biased

Response: We have removed the AUC for training data from the main text, and moved it to the supplement.

- Figure 3: remove training set from figure

Response: We have removed the results for the training set from the main text, and moved them to the supplement.

- Line 100: if the 5 humans SNPs are not significant, why include them in the risk model?

Response: We thank the reviewer for this question. We have revised the model by including only the statistically significant HLA SNPs. Although the two human HLA SNPs have a small absolute effect on NPC risk, their contribution to risk prediction is statistically significant.

- Lines 120-121: how was “did not contribute significantly to accuracy of risk” evaluated/quantified?

Response: We thank the reviewer for this question. We used the AUC in the ROC analyses and NRI to assess the contribution of human HLA SNPs (as well as other modeled factors) to risk prediction. Although the two human HLA SNPs were not the major contributors, they did contribute statistically significantly to the accuracy (**Supplementary Tables 3 and 4**) and goodness-of-fit of the prediction model (**Supplementary Figure 2B and Appended Figure 2**). We have corrected the statement in the revised manuscript (Lines 166-175).

- Line 133: to talk about event rates is misleading: Figure 1B in Supplement does not assess prediction in the population, as only the proportions of cases in a case control study in deciles of the score are evaluated, i.e. the Hosmer Lemeshow test is assessing if the relative risks in deciles agree between the training and validation population. This is only assessing the goodness of fit of the relative risk model and is not the same as saying something about the actually fit of an absolute risk model.

Response: We agree with the reviewer's comments. We have modified the figure legends (**Figure 2B** and **Supplementary Figure 2B**) and the corresponding description in the main text (Lines 173-175) to correctly explain the results regarding the relative risk assessment.

- Line 154: estimated risks in the training set need to be removed.

Response: We have removed the results for the training set from the main text, and moved them to the supplement.

- Line 158: the datasets should not be combined.

Response: We have removed the results for the combined dataset from the revised manuscript.

- Table 2: please compute all the performance characteristics using only the validation data, not the combined dataset

Response: We have removed the results for the combined dataset from the revised manuscript.

- Line 219: if 20% of the population at highest NPC risk need to be screened further, that is a large number. How many cases will be detected in those 20% at highest risk based on the CRS prediction?

Response: We thank the reviewer for this question, which has helped us to improve the presentation in the new manuscript. Approximately 50% of cases can be detected in the top 20% of the highest risk in the validation dataset. Our study suggests that the CRS can be used as the first step of population screening to identify individuals at the highest 20% of NPC risk. These at-risk individuals can then take EBV serology tests as the second step of NPC screening. Of the general population from endemic areas of southern China, 1-5% test

at high risk using EBV VCA-/EBNA-1-IgA serology test⁶. Since CRSs did not show a correlation with EBV serology (**Supplementary Figure 5**), we estimate that about 1-5% of the population at the top 20% NPC risk defined by CRS would test at high risk with EBV serology tests. Using this two-step screening strategy, we estimate that only 0.2-1% of the general adult population will be stratified into the high-risk group. This 0.2-1% of the general population at highest risk may potentially benefit from further preventive actions, for example, routine EBV serology tests and medical imaging-based screening for the early diagnosis and prevention of NPC.

Hence, a two-step screening strategy that combines CRS and EBV serology tests would further concentrate the population at NPC risk efficiently, compared with the CRS test alone. In the revised manuscript, we have amended the discussion and presentation on this two-step screening strategy (Lines 270-284).

- I do not understand the computation of the absolute risk in the supplement, please clarify
Response: We thank the reviewer for pointing out this problem. We have clarified the calculation of the absolute risk among men and women and revised the results (**Figure 4** and **Supplementary Figure 4**) and methods (Lines 382-402) accordingly in the revised manuscript.

1. Hadinoto V, Shapiro M, Sun CC, Thorley-Lawson DA. The Dynamics of EBV Shedding Implicate a Central Role for Epithelial Cells in Amplifying Viral Output. *PLoS Pathog* **5**, e1000496 (2009).
2. Xu F-H, *et al.* An Epidemiological and Molecular Study of the Relationship Between Smoking, Risk of Nasopharyngeal Carcinoma, and Epstein-Barr Virus Activation. *JNCI: Journal of the National Cancer Institute* **104**, 1396-1410 (2012).
3. Chang ET, *et al.* Active and Passive Smoking and Risk of Nasopharyngeal

Carcinoma: A Population-Based Case-Control Study in Southern China. *American Journal of Epidemiology* **185**, 1272-1280 (2017).

4. Leung S-f, *et al.* Plasma Epstein-Barr viral deoxyribonucleic acid quantitation complements tumor-node-metastasis staging prognostication in nasopharyngeal carcinoma. *Journal of clinical oncology* **24**, 5414-5418 (2006).
5. Chan KCA, *et al.* Analysis of Plasma Epstein-Barr Virus DNA to Screen for Nasopharyngeal Cancer. *New England Journal of Medicine* **377**, 513-522 (2017).
6. Ji MF, *et al.* Incidence and mortality of nasopharyngeal carcinoma: interim analysis of a cluster randomized controlled screening trial (PRO-NPC-001) in southern China. *Annals of Oncology*, (2019).
7. Chien Y-C, *et al.* Serologic Markers of Epstein-Barr Virus Infection and Nasopharyngeal Carcinoma in Taiwanese Men. *New England Journal of Medicine* **345**, 1877-1882 (2001).
8. Li K, Lin G-Z, Shen J-C, Zhou Q. Time trends of nasopharyngeal carcinoma in urban Guangzhou over a 12-year period (2000-2011): declines in both incidence and mortality. *Asian Pac J Cancer Prev* **15**, 9899-9903 (2014).
9. Liu Z, *et al.* Two Epstein-Barr Virus-Related Serologic Antibody Tests in Nasopharyngeal Carcinoma Screening: Results From the Initial Phase of a Cluster Randomized Controlled Trial in Southern China. *American Journal of Epidemiology* **177**, 242-250 (2012).
10. Lam WKJ, *et al.* Sequencing Analysis of Plasma Epstein-Barr Virus DNA Reveals Nasopharyngeal Carcinoma-Associated Single Nucleotide Variant Profiles. *Clinical Chemistry* **66**, 598-605 (2020).

11. Cui Q, *et al.* Nasopharyngeal carcinoma risk prediction via salivary detection of host and Epstein-Barr virus genetic variants. *Oncotarget* **8**, 95066-95074 (2016).
12. Xue W-Q, *et al.* Decreased oral Epstein-Barr virus DNA loads in patients with nasopharyngeal carcinoma in Southern China: A case-control and a family-based study. *Cancer Med* **7**, 3453-3464 (2018).
13. Thorley-Lawson DA. EBV Persistence--Introducing the Virus. *Curr Top Microbiol Immunol* **390**, 151-209 (2015).

REVIEWER COMMENTS

Reviewer #1 (Remarks to the Author):

The authors have addressed my comments.

As this assay has the potential to be an important screening tool, it is important to assess the performance of this assay in early stage tumours. Stage I and II tumours are minority of the patients cohorts at study here. As a final suggestion I'd suggest including a common regarding the limitations of the assays performance in this important patient population.

Reviewer #2 (Remarks to the Author):

The authors have addressed all my comments and suggestion in the revised manuscript except the presence of the high risk EBV SNPs in other endemic regions including Indonesia.

As shown in a report from Farrell's group (J Virol. 2018 Oct 29;92(22):e01132-18), based on the EBER2 variants, the EBV genotypes in NPC patients and controls recruited from Indonesia seem to be different from that from South China. The authors should check the three high-risk EBV variants in NPC/control samples from Indonesia in their report and discuss whether the CRS score can apply to other endemic regions.

Reviewer #3 (Remarks to the Author):

"Comprehensive Risk Scores Based on Epstein-Barr Virus and Genetic Variants for Effective Nasopharyngeal Carcinoma Screening", revision

The authors have been responsive to many of my previous comments and the paper is much improved, but some major concerns remain

Main points:

- Controls were frequency matched to cases on sex and age, thus gender and age effects could not be estimated. The authors now removed age and sex from the prediction model (appropriately so). They now also provided details on the computation of the cumulative risk model. However, the computation of the baseline hazard function in the cumulative risk model (line 395) where the baseline is divided by the mean risk in controls assumes that the controls are a completely random sample from the population. This is not the case and thus the baseline estimate is likely not correct.
- The CRS model only will be useful if it can be shown to be unbiased, i.e. well calibrated in the population, which cannot be done with case-control data. The value of assessing the performance of the CRS in case-control data is extremely limited. Thus Figure 2B will lead most readers to misleading conclusions. This was done in a case controls study hugely enriched by cases. In a general population sample that number would be much smaller as incidence of NPC is so low.

Minor comments:

- Figure 2B is unclear, what model does the "predicted probabilities" plotted on the x-axis come from? CRS?
- Figure 3A is redundant, that information is contained in the ROC curve

Reviewer #1 (Remarks to the Author):

The authors have addressed my comments.

As this assay has the potential to be an important screening tool, it is important to assess the performance of this assay in early stage tumours. Stage I and II tumours are minority of the patients cohorts at study here. As a final suggestion I'd suggest including a common regarding the limitations of the assays performance in this important patient population.

Response: Thank you for the very encouraging comments and we have added the discussion on this limitation in the revised paper as suggested by the reviewer (**Lines 267-271**).

This modification in the main text is shown below:

266 different from that in southern China (**Supplementary Table 6**). Despite the small sample size, their
267 evidence might highlight that our CRS model could be population specific. As this CRS approach has
268 the potential to be an important screening tool, it is important to assess the performance in early-stage
269 tumors. However, because Stage I and II NPC tumors were minority of the patient dataset in this
270 study (**Supplementary Table 7**), we could not evaluate the performance of this CRS approach to
271 identify early-stage NPC. Nevertheless, this study offers a compelling proof of concept that using
272 combined EBV and host genotype data and epidemiological risk factor data to predict NPC risk can
273 strongly augment the efficacy of NPC screening based on EBV serology.

Reviewer #2 (Remarks to the Author):

The authors have addressed all my comments and suggestion in the revised manuscript except the presence of the high risk EBV SNPs in other endemic regions including Indonesia. As shown in a report from Farrell's group (J Virol. 2018 Oct 29;92(22):e01132-18), based on the EBER2 variants, the EBV genotypes in NPC patients and controls recruited from Indonesia seem to be different from that from South China. The authors should check the three high-risk EBV variants in NPC/control samples from Indonesia in their report and discuss whether the CRS score can apply to other endemic regions.

Response: Following reviewer's suggestion, we have reviewed the report from Farrell's group (J Virol, 2018)[1] to check the frequency of three high-risk EBV variants in Indonesian samples. Of the 20 NPC cases and 4 controls from Indonesia, the frequencies of two EBV risk

variants for positions 162215 and 162476 are 95% (19) and 70% (14) among cases, and 100% (4) among controls. The risk variant for position 163364 was only identified among cases (10%, 2). This distribution is different from our data in southern China (**Supplementary Table 6**). Despite the small sample size, their evidence might highlight that the CRS model could be population specific. We thank the reviewer for this comment, and this information has been added in the new Discussion (**Lines 264-267**) and **Supplementary Table 6**.

This modification in the main text is shown below:

263 This study has some limitations. The CRS generated in this study was developed and validated in
264 a southern Chinese population with high NPC incidence. As reported in a study, the distribution of
265 these three risk EBV variants among 20 NPC cases and 4 controls recruited from Indonesia²⁷ is quite
266 different from that in southern China (**Supplementary Table 6**). Despite the small sample size, their
267 evidence might highlight that our CRS model could be population specific. As this CRS approach has
268 the potential to be an important screening tool, it is important to assess the performance in early-stage
269 tumors. However, because Stage I and II NPC tumors were minority of the patient dataset in this

Reviewer #3 (Remarks to the Author):

"Comprehensive Risk Scores Based on Epstein-Barr Virus and Genetic Variants for Effective Nasopharyngeal Carcinoma Screening", revision

The authors have been responsive to many of my previous comments and the paper is much improved, but some major concerns remain

Main points:

- Controls were frequency matched to cases on sex and age, thus gender and age effects could not be estimated. The authors now removed age and sex from the prediction model (appropriately so). They now also provided details on the computation of the cumulative risk model. However, the computation of the baseline hazard function in the cumulative risk model (line 395) where the baseline is divided by the mean risk in controls assumes that the controls are a completely random sample from the population. This is not the case and thus the baseline estimate is likely not correct.

Response: We agree with the reviewer that the calculation of the baseline hazard function in

the cumulative risk model based on case-control design could be suboptimal because our controls were sampling with frequency matched on age and sex of cases. The editor also raised this important point to us, and we agree with the suggestion by removing the estimation for cumulative risk in the revised manuscript. We thank the reviewer for the comment, and to estimate the baseline hazard and to develop a cumulative risk model, we are now conducting a population-based cohort study in southern China.

- The CRS model only will be useful if it can be shown to be unbiased, i.e. well calibrated in the population, which cannot be done with case-control data. The value of assessing the performance of the CRS in case-control data is extremely limited. Thus Figure 2B will lead most readers to misleading conclusions. This was done in a case controls study hugely enriched by cases. In a general population sample that number would be much smaller as incidence of NPC is so low.

Response: We agree with Reviewer 3 that the ideal performance test of a polygenic risk score (PRS; in our paper, the CRS model) is in a prospective, population-based cohort. As noted by Reviewer 3, Figure 2B is not a model calibration at the absolute risk level. In the original context, we used Figure 2B to show a better calibration of the CRS model than the other two models in the current case-control dataset. Following your comment, we have addressed and clarified this point in the revised main text (**Lines 173-174**). In addition, we have also moved Figure 2B to Supplementary Figure 2B to avoid potential misunderstandings.

This modification in the main text is shown below:

173 0.770; **Table 2**). Hosmer-Lemeshow plots indicated a better calibration of the comprehensive model
174 than the other two models (Nagelkerke pseudo-R² = 0.23, *P* value of Pearson chi-squared goodness-
175 of-fit test = 0.68, **Supplementary Figure 2B**).

Since the editor has also communicated with us on this point, we thus have attached our communication with the editor for your information:

Reply to the editor: Because specific cancer types, including NPC, have a relatively low incidence in the general population (compared with some common diseases such as cardiovascular diseases and hypertension), demonstrating the performance of any cancer

PRS within the general population—albeit informative—is very challenging, as pointed out by Reviewer 3. Only if a PRS approach is able to identify a subpopulation at a monogenic level of disease risk, as seen with BRCA1 and BRCA2 mutations for breast cancer, for example, can a population-based prospective screening study be conducted with a PRS[2]. In our study, the CRS model indeed identified a subpopulation at a monogenic level of NPC risk. Importantly, as demonstrated in our independent validation dataset, it outperformed most published cancer PRSs in terms of the magnitude of the association between the score and the disease risk (e.g. PRSs based on common variants for breast cancer[3], prostate cancer[4], colorectal cancer[5] and gastric cancer[6], etc.), providing a promising basis for population-wide screening using the CRS approach, **particularly for populations with the highest incidence of NPC in southern China**. Therefore, our current study would pave the way for a long-term, prospective cohort study of NPC population screening.

Minor comments:

- Figure 2B is unclear, what model does the “predicted probabilities” plotted on the x-axis come from? CRS?

Response: We have added the details of three models and clarified that this figure describes “the observed NPC event rates *versus* the expected event rates as predicted by the three risk prediction models indicated” in the legend of Supplementary Figure 2B in the new manuscript.

- Figure 3A is redundant, that information is contained in the ROC curve

Response: We have moved Figure 3A to the supplement.

1. Correia, S., et al., *Sequence Variation of Epstein-Barr Virus: Viral Types, Geography, Codon Usage, and Diseases*. Journal of Virology, 2018. **92**(22): p. e01132-18.
2. Torkamani, A., N.E. Wineinger, and E.J. Topol, *The personal and clinical utility of polygenic risk scores*. Nature Reviews Genetics, 2018. **19**(9): p. 581-590.
3. Mavaddat, N., et al., *Prediction of breast cancer risk based on profiling with common genetic variants*. JNCI: Journal of the National Cancer Institute, 2015. **107**(5).
4. Seibert, T.M., et al., *Polygenic hazard score to guide screening for aggressive prostate cancer: development and validation in large scale cohorts*. BMJ, 2018. **360**: p. j5757.

5. Hsu, L., et al., *A Model to Determine Colorectal Cancer Risk Using Common Genetic Susceptibility Loci*. *Gastroenterology*, 2015. **148**(7): p. 1330-1339.e14.
6. Jin, G., et al., *Genetic risk, incident gastric cancer, and healthy lifestyle: a meta-analysis of genome-wide association studies and prospective cohort study*. *The Lancet Oncology*, 2020. **21**(10): p. 1378-1386.

REVIEWERS' COMMENTS

Reviewer #2 (Remarks to the Author):

The authors have addressed my comments.

Reviewer #3 (Remarks to the Author):

The authors have now corrected remaining issues with the risk prediction model and I have no further comments.

Ruth Pfeiffer

Point-by-point response to the reviewers' comments

Reviewer #2 (Remarks to the Author):

The authors have addressed my comments.

Response: Thank you.

Reviewer #3 (Remarks to the Author):

The authors have now corrected remaining issues with the risk prediction model and I have no further comments.

Response: Thank you.